



# Measurement report: Effect of wind shear on PM10 concentration vertical structure in urban boundary layer in a complex terrain

Piotr Sekuła[1,2], Anita Bokwa[3], Jakub Bartyzel[1], Bogdan Bochenek[2], Łukasz Chmura[1,2], Michał Gałkowski[1,4], Mirosław Zimnoch[1]

[1] Faculty of Physics and Applied Computer Science, AGH-University of Science and Technology, 19 Reymonta St., 30-059 Kraków, Poland
[2] Institute of Meteorology and Water Management, National Research Institute, Branch of Kraków, 14 Piotra Borowego St., 30-215 Kraków, Poland
[3] Institute of Geography and Spatial Management, Jagiellonian University, 7 Gronostajowa St., 30-387 Kraków, Poland, anita.bokwa@uj.edu.pl
[4] Max Planck Institute for Biogeochemistry, Department of Biogeochemical Signals, Hans-Knoll-Str. 10, 07745 Jena, Germany

Correspondence to: Anita Bokwa (anita.bokwa@uj.edu.pl)

**Abstract.** The paper shows wind shear impact on PM10 vertical profiles, in Kraków, southern Poland. The data used consist of background data for two cold seasons (Sep. 2018 to Apr. 2019, and Sep. 2019 to Apr. 2020), and data for several case studies from November 2019 to March 2020. The data is composed of PM10 measurements, model data, and wind speed and direction data. The background model data come from operational forecast results of AROME model. PM10 concentration in the vertical profile was measured with a sightseeing balloon. Significant spatial variability of wind field was found. The case studies represent the conditions with much lower wind speed and a much higher PM10 levels than the seasonal average. The inversions were much more frequent than on average, too. Wind shear turned out to be the most important factor in terms of PM10 vertical profile modification. It is generated due to the relief impact, i.e. the presence of a large valley, blocked on one side with the hills. The analysis of PM10 profiles from all flights allows to distinguish three vertical zones of potential air pollution hazard within the valley (about 100 m deep) and the city of Kraków: 1. up to about 60 m a.g.l. – the zone where during periods of low wind speed, air pollution is potentially the highest and the duration of such high levels is the longest, i.e. the zone with the worst aerosanitary conditions; 2. about 60-100 m a.g.l. – transitional zone where the large decrease of PM10 levels with height is observed; 3. above 100-120 m a.g.l. – the zone where air quality is significantly better than in the zone 1, either due to the increase of the wind speed, or due to the wind direction change and advection of different, clean air masses.

## 1 Introduction

Particulate matter (PM) concentration remains one of the most relevant air-quality concerns in urban environments (Thürkow et al., 2021). Exposure to ambient PM concentration with diameter below 10 µm (PM10) can cause lung irritation, cellular damage, coughing asthma, and cardiovascular diseases (Jeong, 2013). Particles with diameter below 1 µm (i.e. fine and ultrafine particles which constitute in most cases the majority of PM10 fraction) have the strongest impact on health because they can reach the deepest portions of the airways or even the blood stream (Franchini and Mannucci, 2007, 2011). Presence of the particulate matter in the ambient air is



the result of multiple physio-chemical processes, including local emission, chemical transformation, long-range
transport, vertical mixing and deposition, most of which are dependent on meteorological conditions across a large
range of spatial and temporal scales (Zhang et al., 2015; Zhou et al., 2020; Thürkow et al., 2021).
Local meteorological conditions determine primarily the dispersion of air pollutants, their removal (Trompetter et
al., 2013), but they also affect chemical and physical process linked to the origin of the primary and secondary
aerosols (Zhou et al., 2020). One of the mostly studied meteorological phenomena is the occurrence of above-
ground air temperature gradient inversion, which has a direct impact on the vertical distribution of the
concentration of $PM_{10}$ and its individual components, e.g. black carbon (Zhou et al., 2020) or organic $PM_{10}$ tracers
like levoglucosan (Marynowski et al., 2020). Numerous studies indicate that an important factor that affects the
pollution profile is the wind profile (Li and Han, 2016; Zhou et al., 2020), occurrence of low-level jet (Li et al.,
2012; Li et al., 2019) or downward flows of pollutants (Han et al., 2018) which may strongly modify diurnal cycle
of a pollutant concentration at the lowest part of the troposphere.
The vertical structure of the pollutant concentration strongly depends on many factors, including season,
meteorological conditions (Wang et al., 2018), topography (Trompetter et al., 2013; Strbova et al., 2017), seasonal
variability of local emissions and long-range transport (Li and Han, 2016). Due to this fact it is necessary to
continuously study the spatial and vertical distribution of air pollution concentration in urbanized areas to better
determine its sources and processes leading to abundant air pollution.
Research on the vertical structure of air pollution has been carried out in the past using several methods: stationary
point measurements in the profile using the available infrastructure (e.g. Marynowski et al., 2020), balloon flights
(e.g. Han et al., 2018; Renard et al., 2020), by airplane or unmanned aerial vehicle (UAV) (Liu et al., 2020),
LIDAR (Li and Han, 2016; Wang et al., 2020) or with the use of satellite data (Ferrero et al., 2019). The highest
vertical resolution can be achieved with the use of an aircraft (plane, balloon, UAV), however these methods have
certain limitations, e.g. lifting capacity, limited flight time and limited maximum reachable altitude, and they
cannot operate during unfavorable weather conditions.
Throughout the previously published studies focused on the topic of lower-tropospheric air pollution, several types
of the pollution concentration vertical profiles can be distinguished:
- two layers with significantly different concentration, i.e. high concentration in the stratum from the ground level
to a certain height, then a transition layer with a rapid decrease in pollutant concentration with height, and a stratum
with a low concentration in the profile above; usually linked to thermal inversion occurrence (Strbova et al., 2017;
Wang et al., 2018; Samad et al., 2020);
- a large, constant decrease of a pollutant concentration with height, resulting e.g. from a strong surface emission
of a pollutant during stable conditions, or from katabatic flows bringing the pollutants (Strbova et al., 2017), and
from removal of the pollutants from upper layers;
- the occurrence of a layer with increased concentration of air pollution at a certain height, connected to vertical
diffusion (Strbova et al., 2017) or diffusion of plumes from elevated sources (Xu et al., 2019);
- a slight decrease of air pollution with height connected to the occurrence of strong vertical movements (Strbova
et al., 2017) or removal of local air pollution due to synoptic processes linked to the advection of air masses.
It is noteworthy that many recent studies of air pollution concentration's vertical structure in cities were realized
mainly for areas with little variation in the topography (e.g. Paris (Renard et al., 2020), Tianjin (Han et al., 2018)),
including coastal areas (Guangzhou (Zhou et al., 2020), Shanghai (Zhang et al., 2017)). In fact, the understanding,



and the quantification of pollutant dispersion over complex terrain are much more difficult than over flat areas, as
dispersion processes are affected by atmospheric interactions with the orography at different spatial scales
(Giovannini et al., 2020). Studies presenting vertical profiles of pollutants' concentration in urbanized valleys are
still necessary to better understand impact of meteorology and topography on air pollutant dispersion (Strbova et
al., 2017; Zhao et al., 2019; Samad et al., 2020).
A key parameter affecting pollutant concentration during the daytime is the height of the atmospheric boundary
layer (ABL), which determines the volume of atmosphere available for pollutant dispersion. Turbulent mixing is
a key factor which controls the evolution of the ABL depth (Giovannini et al., 2020). One of the important factors
is the wind shear as it may essentially modify the structure of mean flow and turbulence in the convective boundary
layer (CBL), e.g. by stretching and decoupling of the turbulent structures production or separation of a single-layer
CBL into two-layer structure (Fedorovich and Conzemius, 2008; Rodier et al., 2017). Studies presenting the impact
of ABL dynamics on vertical pollutant structure indicate that a low-level jet combined with a strong wind shear
affect the transportation of the pollution e.g. by removing it (Trompetter et al., 2013) or bringing it in (pushing
into the residual layer), and by favoring the growth of ABL height and weakening the stability of the atmosphere
(Li et al., 2019).
The present study is focused on the impact of wind shear on the vertical profile of $PM_{10}$ concentration in Kraków,
southern Poland, a city located in a large valley. The properties of ABL, including vertical profile of wind speed
and direction, are strongly modified both by the relief and the synoptic situation, and so are the air pollution's
dispersion conditions which in turn affects the pollutants concentration's profile. Those circumstances are of the
highest importance in a city located in a valley as the built-up areas are located both in the valley bottom as well
as on its slopes, i.e. in a vertical profile of the landform. Kraków is a good study area for such considerations as it
is located in diversified environmental conditions (described in detail in section 2), and despite various legal
actions aimed to reduce local $PM_{10}$ emissions, daily limit values for $PM_{10}$ are still exceeded during cold seasons.
Moreover, Kraków is representative for many cities located in Central Europe where aerosanitary conditions are
relatively worse in comparison to the cities in the western part of the continent, as presented e.g. in the reports of
the European Environment Agency (Air… 2020). Poor air quality is, on one hand, the result of PM10 emissions
which in the case of Poland are among the highest in Europe (PM10 emissions… 2020), however, with a
decreasing trend in recent years (Raport… 2017). But high PM10 concentrations are also linked to a long-range
transport of air pollution from other countries (Godłowska et al. 2015). In the Lesser Poland region (*Małopolska)*
where Kraków is located, the main source of PM10 is the emission from the municipal and housing *sec*tor (78.9%
of the annual emission), from transportation (5%), and from industry (7.8%). In Kraków, the emissions related to
vehicle traffic account for as much as 12% of annual emission (Raport…2020). Understanding the meteorological
processes leading to the enhanced concentration levels is one of the key factors to enable the development
strategies for inhabited areas to further reduce the number of smog episodes. To date no studies presenting temporal
variability of $PM_{10}$ concentration in vertical profile in cold season has been reported in that region.
**2   Study area**
Kraków is a large valley city located in the Wisła River valley, which is parallel to the border of the Carpathian
Mts. to the south, and the border of Polish Uplands to the north (Fig. 1). About 100 km south of Kraków, there is
the highest ridge of the Carpathians, the Tatra Mts. Kraków is the second largest city of Poland, located in the
Lesser Poland region (*Małopolska*), with an area of 326.8 km² and the official number of inhabitants reaching 771



thousand (as of Dec. 2018 (Kraków, 2019)). Kraków agglomeration consists of the city itself and highly populated
towns and villages which surround it, with the total number of inhabitants is estimated to exceed 1 million. The
city's area belongs to three different geographical regions and geological structures, i.e. the Polish Uplands, the
Western Carpathians, and the basins of the Carpathian Foredeep in between (Bokwa, 2009). The central part of
the city is located in the Wisła River valley, at an altitude of about 200 m a.s.l. In the western part of Kraków, the
valley is as narrow as 1 km. However, in the eastern part of the city, the valley widens to about 10 km and there is
a system of river terraces. East of the city's borders, the Raba River enters the Wisła River with a valley cutting
the Carpathian Foothills from the south to the north. The hilltops bordering the city to the north and the south reach
about 100 m above the river valley floor, similar to the hilltops in the western part of the valley which means that
the city is located in a semi-concave land form (open only to the east), and sheltered from the prevailing western
winds (Fig. 1). The local scale processes linked to the impact of relief include, for example, katabatic flows, cold
air pool (CAP) formation, frequent air temperature inversions, much lower wind speed in the valley floor than at
the hilltops (e.g. Hess, 1974). According to the studies on thermal stratification obtained for Kraków by using
sodar measurements with hourly resolution, in the months from October to March, the mean monthly frequency
of stable atmosphere conditions varies from 58.1 % in March to 74.0 % in December (Godłowska, 2019). All
factors mentioned above contribute to the poor natural ventilation of the city and the occurrence of high $PM_{10}$
levels, especially in the heating season.

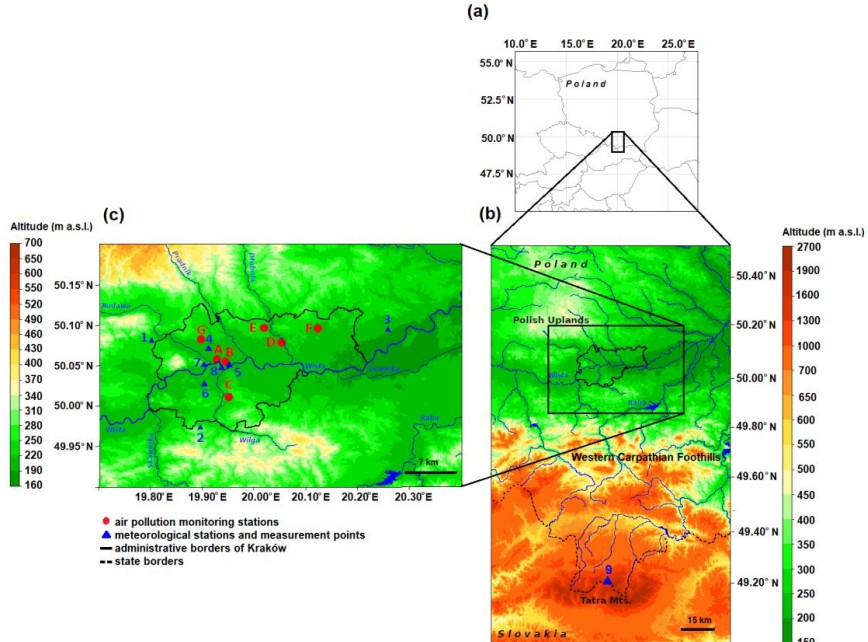


*Figure 1. Location of the region studied: a. in Central Europe, b. in southern Poland, c. at the junction of the*
*Wisła River valley, Polish Uplands and the Western Carpathian Foothills.*
Explanations: numbers and letters as in table 1 and 2





**3    Data and methods**
**3.1 Surface measurements**
The data used consist of background data for two cold seasons (Sep. 2018 to Apr. 2019, and Sep. 2019 to
Apr. 2020), and data for several case studies from November 2019 to March 2020. The background data is
composed of $PM_{10}$ measurements from 7 stations, model data, and wind speed and direction data from 4
meteorological stations. The data for case studies come from 7 stations with $PM_{10}$ measurements, model analyses,
and 8 meteorological stations (wind speed and direction, air temperature, air humidity and cloudiness) (Fig. 1,
Table 1 and 2).
Data on $PM_{10}$ concentrations in Kraków come from data bases of the National Inspectorate of
Environmental Protection (NIEP) (https://powietrze.gios.gov.pl/pjp/archives). Mean hourly data from 7
measurement points were used (Table 1). The measurement points represent several parts of the city, and are
located in various types of landform and land use/land cover (see Fig. 1 for the location of the measurement points):
A.  Krasińskiego St.: street canyon in the city center, in the bottom of the Wisła River valley, with a very
busy municipal transportation route and intensive traffic;
B.  Dietla St.: a busy cross-road in the city center, in the bottom of the Wisła River valley, with intensive
tram, bus and car traffic;
C.  Kurdwanów district: suburban area with a large district of blocks of flats, in the southern part of the city,
about 50 m above the valley floor;
D.  Bulwarowa St.: suburban area with a large district of blocks of flats, located close to the steelworks, in
the eastern part of the city, at a terrace of the Wisła River;
E.  Piastów district: suburban area with a large district of blocks of flats, in the eastern part of the city, on the
upland slope, about 50 m above the valley floor;
F.  Wadów district: suburban area with agriculture activity and loose residential built-up, located close to the
steelworks, at a river terrace in the eastern part of the Wisła valley;
G.  Złoty Róg St.: suburban area with a large district of blocks of flats and residential built-up, on the upland
slope, in the western part of the city.

Table 1. Location of air pollution monitoring stations in Kraków

| Symbol | Station | Lat N | Lon E | Altitude (m a.s.l.) | Land form |
|--------|---------|-------|-------|---------------------|-----------|
| A | Krasińskiego St | 50.06 | 19.93 | 207 | Valley bottom |
| B | Dietla St. | 50.05 | 19.94 | 209 | Valley bottom |
| C | Kurdwanów district | 50.01 | 19.95 | 223 | Valley slope |
| D | Bulwarowa St. | 50.08 | 20.05 | 195 | Valley bottom |
| E | Piastów district | 50.10 | 20.02 | 239 | Valley slope |
| F | Wadów district | 50.10 | 20.12 | 218 | Valley bottom |
| G | Złoty Róg St. | 50.08 | 19.90 | 218 | Valley slope |

Background data on wind conditions in the Wisła river valley and the neighboring hilltop were obtained from the
stations of the Institute of Meteorology and Water Management – National Research Institute (IMWM-NRI)
(Balice, Igołomia and Libertów) and the station of AGH University of Science and Technology (AGH UST),
located in the Reymonta St. (city center), on the roof of the Faculty of Physics and Applied Computer Science.



Wind speed and direction data of hourly resolution were used. Table 2 and figure 1 show the location of the stations
and the range of measurements.
**3.2 Modelling systems**
The Aire Limitée Adaptation Dynamique Développement International (ALADIN) system is a numerical
weather prediction (NWP) system developed by the international ALADIN consortium for operational weather
forecasting and research purposes (Termonia et al., 2018). One of the consortium's development work is to provide
several configurations of limited-area model (LAM), which were precisely validated to be used for operational
weather forecasting at the 16 partner institutes. These configurations are called the ALADIN canonical model
configurations (CMCs). Currently there are three canonical model configurations: 1. ALADIN baseline CMC, 2.
Application of Research to Operations at Mesoscale (AROME) CMC, and 3. ALADIN–AROME (ALARO) CMC.
AROME CMC and ALARO CMC are operationally used in IMWM-NRI, together with the CY43T2.
The background model data come from operational forecast results of AROME model.  Operational
model AROME CMC 2 km has a horizontal resolution of 2 km x 2 km and 70 vertical levels, the forecast length
is 30 h. Size of AROME CMC 2 km domain is 799 x 799 points with centered on geographical point 19.3°E
52.3°N. The location of the lowest model level is at 9 m above ground level, and the model top is located at 65 km
above ground level. During the analyzed periods model version has been changed from CY40T1 to CY43T2 (11
Feb. 2020). Seasonal verification of the AROME CMC model forecast results showed compliance new version
with the previous one (Bochenek et al., 2020).
ALARO model was used to prepare lateral boundary data for AROME model.  ALARO CMC CY43T2
is a non-hydrostatic model, with a horizontal resolution of 4 x 4 km and 70 vertical levels. The model configuration
ALARO CMC and AROME CMC has been validated by the ALADIN team at IMWM-NRI for CY43T2 for
resolution 4 km x 4 km and 2 km x 2 km, respectively.
Archival forecasts of the AROME CMC model with temporal resolution of 1h (forecast hours from 6th
to 29th), were used to study the characteristics of vertical wind and temperature profiles in the valley, with special
focus on 3 height levels (50, 100 and 200 m a.g.l.), as the valley depth is about 100 m. Analyses were conducted
at 4 selected points, representing Balice meteorological station, TV tower with vertical profile measurements, city
center, and Bulwarowa St. ($PM_{10}$ measurements). The points mentioned are located along the valley bottom in the
W-E cross section.
**3.3 Vertical profile observations and data verification**
For the period from November 2019 to March 2020, additional data for the case studies are available.
They consist of measurements of $PM_{10}$ concentration in the vertical profile, performed on 31 days selected. The
$PM_{10}$ profiles' measurements were carried out in cooperation with the company Balon Widokowy sp. z o. o.
(http://balonwidokowy.pl/ ) which operates commercially the sightseeing balloon in Kraków. The $PM_{10}$
measurements were conducted up to maximum altitude of almost 300 m a.g.l. Balloon flights were performed in
the western part of the city, at the Wisła River, in the city center, close to the air quality monitoring stations
Krasińskiego St. and Dietla St.
Measurements of $PM_{10}$ concentration in the vertical profile were conducted by Personal Dust Monitor (PoDust
v1.1) system based on low-cost Plantower PMS1003 optical dust sensor and Arduino platform presented on Figure
2. The measurement system was attached to outside of the balloon basket. It was build based on the Arduino Mega
2560 microcontroller, responsible for communication with the sensors, storing the measurements with 1 second



resolution on the memory card, and sending information in real time to the database using WiFi connection. To
reduce the impact of water vapor on $PM_{10}$ measurement during the fog conditions, sensor inlet was heated up to
60ºC. To provide information on an actual location and other environmental conditions, the system was equipped
with a GPS receiver and thermo/hygro/baro sensor providing e.g. the altitude estimated with combined GPS and
barometer signals.
The measurement campaign covered the period from November 28, 2019, to March 3, 2020, during which 317
flights were conducted (31 days, 634 vertical profiles). Maximum flight altitude varied between 78 and 284 m
a.g.l., depending on vertical wind profile and number of passengers. Typical flight altitude during sightseeing
flight was 150 m a.g.l., but during low wind speed at higher altitudes and low passenger load, the maximum altitude
was increased. The measurements were performed at different hours. The balloon's flight speed does not exceed 1
$m \cdot s^{-1}$ (ascent up to 0.8 $m \cdot s^{-1}$, descent approximately up to 0.6 $m \cdot s^{-1}$), flight time (ascent/descent) depended on the
maximum altitude and ranged from 2-3 minutes (for maximum height 100 m a.g.l.) up to 6-10 minutes (for
maximum height 300 m a.g.l.).
The frequency of flights depended on meteorological conditions and the number of customers. More than 70% of
the flights were performed up to 180 m above ground level, flights reaching over 200 m above ground level made
only 15% of cases. Almost 50% of vertical profiles were conducted between 12 and 15 UTC, while profiles from
15 to 20 UTC constitute 23% of cases. The flight altitude depended on the wind speed in the whole vertical profile
of the balloon range, which was measured directly during the flight. Figure 2 presents comparison of $PM_{10}$
measurements from balloon device, conducted at 2 m a.g.l., and measurements from the nearby Krasińskiego
station. As the measurements from Krasińskiego station are of hourly resolution, linear interpolation of two
adjacent measurements was applied to obtain the same data resolution as for the balloon. The intersection point of
the straight line matching the graph has been set to 0 because tests on the Plantower sensor have shown the correct
measurement for a concentration close to 0 $\mu g \cdot m^{-3}$.

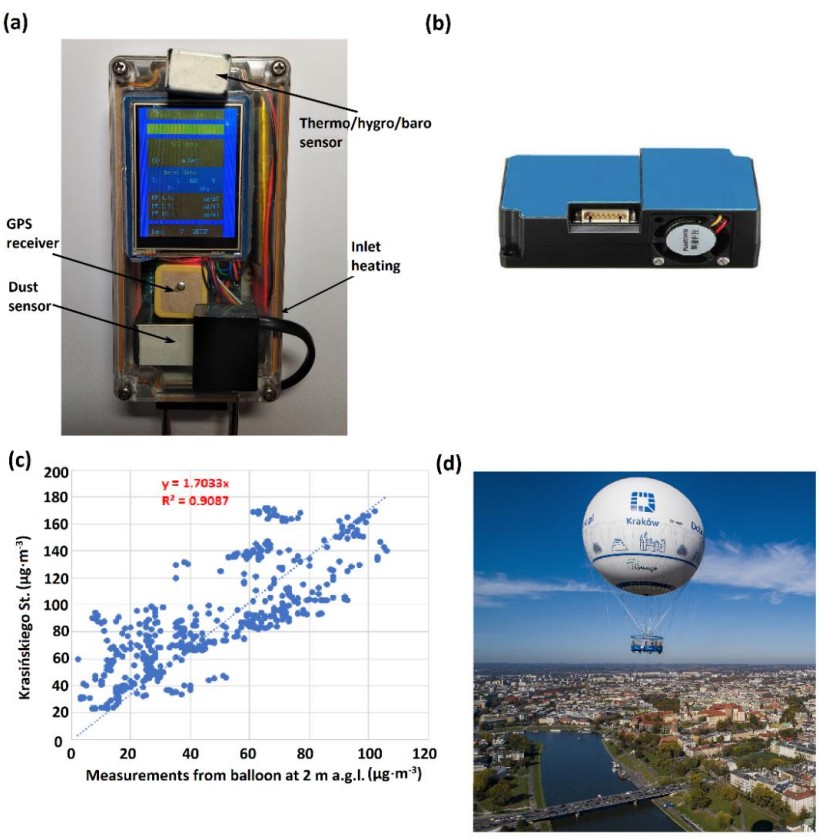


*Figure 2. Self-designed and bulit air pollution measuring system (a), low cost sensor Plantower PMS1003 PM component (b), correlation of measurements from balloon location and closest air pollution station (Krasińskiego St.) with fitted regression curve and R squared factor (c) and (d) sightseeing balloon (source: http://balonwidokowy.pl).*

Data on meteorological conditions, in synoptic and local scale, for Kraków for days with balloon flights, were obtained from the meteorological stations already mentioned above, and additionally from two stations administered by the Jagiellonian University (JU) (Campus JU, Botanical Gardens) and one station administered by IMWM-NRI (Kasprowy Wierch Mt., in the Tatra Mts.).  The JU also administers measurements at the television tower (the technical details can be found in Bokwa A. 2010); the tower belongs to Emitel company.

 Due to possible effect of foehn occurrence on ABL modification, potential foehn occurrence was determined based on the criteria of Ustrnul (1992), upon the analysis of the measurement data from the synoptic stations Kasprowy Wierch Mt. (wind speed and direction) and Balice (wind speed and direction, and air humidity). One of the criteria determining foehn occurrence in Kraków is the presence of Altocumulus lenticularis clouds (Ac len) which are one of the effects of mountain waves. Information about Ac len clouds occurrence was obtained from the station in the Botanical Gardens in Krakow. Data on air temperature in the vertical profile of the Wisła river valley were obtained from stationary measurements at the altitudes 2, 50 and 100 m a.g.l., from TV tower located



in the western part of the valley. Table 2 and figure 1 show the location of the stations and the range of
measurements.
*Table 2. Location of meteorological stations in Kraków and its vicinities, station Kasprowy Wierch, balloon*
*measurement point and meteorological elements used in the study.*

| No. | Station | Lat N | Lon E | Altitude (m a.s.l.) | Manager of the station | Land form | Elements Used |
|---|---|---|---|---|---|---|---|
| 1 | Balice | 50.08 | 19.80 | 237 | IMWM-NRI | Valley bottom | V, D, T, RH |
| 2 | Libertów | 49.97 | 19.90 | 314 | IMWM-NRI | Hill top | V, D, T, RH |
| 3 | Igołomia | 50.09 | 20.26 | 202 | IMWM-NRI | Valley bottom | V, D, T, RH |
| 4 | Reymonta St. | 50.07 | 19.91 | 220 | AGH UST | Valley bottom | V, D, T, RH |
| 5 | Botanical Gardens | 50.05 | 19.95 | 206 | JU | Valley bottom | V, D, Ac len clouds |
| 6 | Campus JU | 50.03 | 19.90 | 233 | JU | Valley bottom | V, D |
| 7 | TV Tower: 2 m a.g.l. 50 m a.g.l. 100 m a.g.l. | 50.05 | 19.90 | 222 272 322 | JU | Valley bottom | T, RH |
| 8 | Balloon measurement point | 50.05 | 19.94 | 200 | AGH UST | Valley bottom | PM10 |
| 9 | Kasprowy Wierch | 49.23 | 19.98 | 1998 | IMWM-NRI | Mountain peak | V, D, T, RH |

Explanations: AGH UST – AGH University of Science and Technology, JU – Jagiellonian University.
More information about the measurement points administered by JU can be found in Bokwa (2010). V –
wind speed, D – wind direction, T – air temperature, RH – relative humidity

For the analysis of case studies data, a different model configuration was used than for background data from the
two cold seasons. Nonoperational configuration of the AROME CMC 1 km x 1 km CY43T2 (AROME CMC 1
km) was applied. Operational model ALARO CY43T2 was used to prepare lateral boundary data for AROME
model version CY43T2. Non-hydrostatic model AROME CMC 1 km has a horizontal resolution of 1 km x 1 km
and 87 vertical levels, the forecast length was 30 h. Size of AROME CMC 1 km domain was 810 x 810 points
with centered on geographical point 20°E 50°N. The location of the lowest model level is at 9 m above ground
level, and the model top is located at 50 km above ground level. Details concerning the height of the lowest model
levels up to 3 km altitude, information about parametrization schemes used in AROME model and topographic
map of model domain are included in table A1, A2 and fig. A1. The data obtained with the model were used to
provide vertical profiles of wind speed and direction, air temperature, relative humidity and Turbulent Kinetic
Energy (TKE) with 1-hour temporal resolution, in the points representative for a western, central and eastern part
of the city, corresponding to the measurements in Balice, Bulwarowa St. and balloon measurement point
respectively. Additionally, N-S cross-sections through the valley at those points were obtained for the same
elements. For selected cases, wind, TKE and air temperature fields at selected levels were obtained for the whole
area of Kraków and its surroundings.





Verification of forecast results of AROME CMC 1km was performed for 24-h periods (i.e., from 6th to 29th hour
of forecast with 1-hour resolution) for selected 31 days of the case study period. Data obtained from 4
meteorological stations (Balice, Libertów, Igołomia and Reymonta St.) were used to verify the model forecast of
air temperature, air humidity and wind components in the valley bottom and at the hill top. The value of root mean
square error (RMSE) between observation and forecast were lower than 2°C for air temperature, 1.5 m·s$^{-1}$ for wind
speed and 14% for relative humidity at all meteorological stations. Air temperature and humidity measurements
at 50 and 100 m a.g.l. from TV tower station were used to verify model forecast of atmosphere stratification in the
west part of the Wisła River valley. Values of RMSE and difference (bias) for air temperature and humidity for
both altitudes (i.e. 50 and 100 m) are similar, on average RMSE was equal 1.5°C for air temperature and 9.5% for
relative humidity.
Data analysis for background period (i.e. two cold seasons) included calculation of standard characteristics for
particular elements studied, in order to: 1. determine their spatial variability in the study area; 2. define wind shear
conditions; and 3. in order to be used further for the verification of the representativeness of the case study period.
The indices used included wind roses for the ground stations, wind speed histograms for three levels (50, 100 and
200 m a.g.l.), air temperature gradients, differences in PM$_{10}$ concentrations between the stations, and the
correlation between PM$_{10}$ concentrations and wind speed.
For the case study period, first the PM$_{10}$ concentration vertical profiles were classified with a subjective method of
fitting the linear curve to each vertical profile. Based on R squared coefficient, the angle of the straight line and
residual values classification has been made. Each profile was checked manually whether it was correctly assigned
to a given group. For this purpose, neighboring flights on a given day were analyzed, too. Objective classification
methods could not be used due to differences in flight heights and the PM$_{10}$ measurement altitudes in particular
flights. Three groups/patterns of PM$_{10}$ concentration vertical profiles were obtained, and for each of them all
meteorological data were analyzed in order to determine their significance in controlling the air pollution vertical
structure.
**4    Results**
**4.1 Spatial and temporal variability of anemological conditions**
Analysis of the data on wind speed and direction from three meteorological stations in the Wisła valley (Balice,
Reymonta St., Igołomia) and one station in the nearby hilltop (Libertów) for the two cold seasons (Sep. 2018 to
Apr. 2019 and from Sep. 2019 to Apr. 2020) indicated significant spatial variability of that element due to the
complexity of the landforms and the presence of urban structures. However, the differences of the wind structure
between the both seasons were negligible. In terms of spatial variability, the average frequency of weak wind (up
to 2 m·s$^{-1}$) varied from 43% in Balice to 61% in Reymonta St.; in Libertów and Igołomia the values reached 50%
and 53%, respectively. For the wind speed ≥4 m·s$^{-1}$, the highest average frequency was measured in Balice (27%),
while in Libertów and Reymonta St. it did not exceed 10%, and in Igołomia reached 21%. Wind speed ≥10 m·s$^{-1}$,
was noted in Igołomia and Balice only. Dominant wind directions are strongly linked to the relief impact. In Balice
those are SW and NE, in Igołomia and Reymonta St. W and E, while in Libertów it is the western sector: SW to
WNW (Fig. A2).



Similar calculations were also performed for the case studies period, i.e. 31 days during which the flights were
conducted, within the period from November 28, 2019 to March 3, 2020, in order to check whether these results
can be treated as representative for the whole cold period. The frequency of wind speed $\leq 2$ m·s$^{-1}$ was much larger
than the average value for both seasons: from 62% in Balice to 83% in Reymonta St., while the frequency of wind
speed $\geq 4$ m·s$^{-1}$ was much smaller: from 0.1% in Reymonta St. to 7.9% in Balice. Dominant wind directions for
the case study period did not differ significantly from the average values for both seasons. Therefore, the case
studies period can be considered as representing days with very low wind speed at the station level.
On the basis of archival forecasts of the AROME operational model, the characteristics of vertical wind profiles
in the valley for four points located in the valley bottom in a W-E cross-section (i.e. Balice, TV tower, city center,
Bulwarowa St.), for the two seasons, were examined at three levels: 50, 100 and 200 m a.g.l. and for every hour
of the day. The analysis did not show significant differences between the seasons. For nearly 50% of the cases, the
velocity at 50 m a.g.l. in the valley did not exceed 4 m·s$^{-1}$. Wind speed at levels 100 and 200 m a.g.l. did not exceed
10 m·s$^{-1}$ and 12 m·s$^{-1}$ for more than 90% of cases, respectively.
Wind direction forecasts at the three levels were used to analyze the frequency of significant wind direction change
in the vertical profile (wind shear), between levels 50 and 100 m a.g.l., 100 and 200 m a.g.l. and 50 and 200 m
a.g.l. Minimum value of significant wind direction change was set to 20°, on the basis of analyses. Wind direction
studies were performed for diurnal (i.e. 6 to 17 UTC) and nocturnal (i.e. 18 to 5 UTC) periods. For the point
representing city center, and located close to the balloon sounding site, for both cold seasons, the percentage of
large wind direction changes which lasted more than 4 hours (between levels 50 and 200 m a.g.l.) equaled 9.5%
and 31.9% during daytime and nighttime, respectively. The values for the case study period reached 42% and 52%,
and for the changes which lasted over 4 hours it was 23.7% and 46.2%.
On the basis of the above comparisons, it is possible to conclude that on the days which belong to the case study
period, wind speed was much lower than on average during both cold seasons, while large wind direction changes
were much more frequent.
**4.2 Spatial and temporal PM10 concentrations' variability**
The analysis of data on $PM_{10}$ concentration from all monitoring points operated by NIEP and described in section
3, from both cold periods analyzed, was performed in order to determine to what extent the measurements of the
$PM_{10}$ vertical profile realized close to the city center, in the western, narrow part of the valley, are representative
for other city's areas. First, significant difference were found between both of the analysed cold seasons; in the
season 2019-2020, the mean concentrations were lower than in the previous cold season at all stations, except
Bulwarowa St. The number of days with mean daily concentration $\leq 50$ μg·m$^{-3}$ increased by as much as 15% in
Kurdwanów dist. and Dietla St., with a simultaneous decrease in the number of days with mean daily concentration
50-100 (-10% on Kurdwanów dist. and -8% on Dietla St.). The number of days with an average daily concentration
$\geq 50$ μg·m$^{-3}$ in the season 2019-2020 ranged between 35 and 63 for most of the stations except the Krasińskiego
St., located close to the balloon site, where the number of such days was equal to 101. In the season 2019-2020,
days with mean daily concentration of 100-150 μg·m$^{-3}$ occurred at four stations only: Krasińskiego St.: 14 days,
Bulwarowa St.: 7 days, Kurdwanów dist.: 4 days, Złoty Róg St.: 3 days, while in 2018-2019, such high
concentrations occurred almost at the same stations, but the numbers were significantly higher, e.g. 28 days in





Krasińskiego St., and from 12 to 14 days in Złoty Róg St., Dietla St., and Kurdwanów dist. Maximum PM10
hourly concentration reached 378 µg·m⁻³ in Dietla St. on 18.02.2019. Therefore, it can be stated that the western
part of the city, located in the narrow part of the valley floor, experiences much worse air pollution concerning
PM10 than the eastern part, located in the wide part of the valley. The vertical $PM_{10}$ measurements can be then
considered representative for the western part of the valley.
As weak winds prevailed during the case study periods, hourly $PM_{10}$ concentrations were analysed for particular
wind speed ranges, and wind measurements from Reymonta St. were used (i.e. representative for the western part
of the city). Concerning high $PM_{10}$ levels, which are the most dangerous for human health, the percentage of the
cases with wind speeds below 1 m·s⁻¹ (during the both cold seasons) when the concentration was higher than 100
µg·m⁻³ varied from 7.3% (Wadów dist.), 10-11% (Dietla St., Bulwarowa St. and Piastów dist.) 13.6% at Złoty Róg
St., to 15.3% at Kurdwanów dist. and 25.7% at Krasińskiego St. For cases ≥ 150 µg·m⁻³, the values varied from
0.7-0.8% (Bulwarowa St., Piastów and Wadów dist.), 1.6% at Dietla St., 1.9% at Złoty Róg St., to 4.1% at
Kurdwanów dist. and 5.7% at Krasińskiego St. The data shows large differences in PM10 horizontal distribution
within the city, and a relatively high frequency of $PM_{10}$ dangerous concentrations, as high as double the allowed
mean daily level.
Figure A3 shows the correlation between $PM_{10}$ concentrations at individual air pollution stations and the wind
speed at Reymonta St. The logarithmic curves were fitted to the data.
Due to the fact that $PM_{10}$ levels differ significantly between the two cold periods analyzed (i.e. 2018-2019 and
2019-2020), $PM_{10}$ data for the case studies period were compared with the data for the whole season 2019-2020
only, in order to check their representativeness for the season. During the case studies period, hourly $PM_{10}$
concentrations ≤ 50 µg·m⁻³ reached from 23% for Krasińskiego St. to 50-60% for the Dietla St., Piastów and
Wadów districts, while during the whole cold season 2019-2020 they were much more frequent and varied from
57% for Krasińskiego St. to over 80% for Dietla St., Piastów and Wadów districts. Parallel, values ≥ 150 µg·m⁻³
for most of the stations were up to 3% (with a minimum in Dietla St. 0.4%) but in Krasinskiego St. they reached
7%, while for the whole season the highest value was 1.3%. That means that the case studies represent not only
the conditions with much lower wind speed than the seasonal average but also the conditions with a much higher
PM10 levels than on average.
**4.3 Vertical air temperature gradient**
Based on the high-resolution forecasts of the AROME CMC 1 km model, an analysis of the vertical temperature
gradient between the model level 50 and 220 m a.g.l. for the city center, for the case studies period, against the
background data from two cold seasons, has been performed. The presence of a thermal inversion is an important
factor which limits the PM10 dispersion conditions, and therefore contributes to its high levels. The gradient values
were calculated separately for the daytime (6-17 UTC) and nighttime (18-5 UTC), as the phenomenon is usually
much more frequent during the nighttime than daytime. The frequency of a gradient greater than 0.5°C/100 m (i.e.
thermal inversion) in the night time was rather similar in the case study period (48%) and in the cold seasons
(38%), while during the daytime, the value for case study period was much larger than for both seasons (32% and
7%, respectively). It means that during the study period, the inversions were much more frequent than on average
in the cold season which contributed to the much higher PM10 concentrations, mentioned above.



The frequency of thermal inversion is linked to wind speed (Table A3). An analysis of the temperature gradient
versus wind speed at 50 m a.g.l. was performed for the both cold seasons, jointly. The studies indicated that for
wind speed $< 2 \mathrm{m \cdot s^{-1}}$ the frequency of the gradient greater than 0.5°C/100 m was 45%, and for wind speed 2-4 m·s⁻
¹ it decreased to 31% of cases. High PM10 concentrations in the study period were then the effect of joint impact
of low wind speed and thermal inversion, generated by the city location in the concave landform.
**4.4 Vertical profiles of PM$_{10}$ concentration**
There were three types of PM$_{10}$ vertical profiles distinguished (Fig. 3):
•   Type I – almost constant value of PM$_{10}$ concentration in the vertical profile (small fluctuations, weak

decrease);

•   Type II – strong decrease of PM$_{10}$ concentration in the vertical profile;
•   Type III - the occurrence of three layers of PM$_{10}$ concentration: 1. constant concentration in the lower

part of the profile, 2. transition layer above, and 3. the upper layer where a sudden drop of PM$_{10}$

concentration is observed.


Out of 31 analyzed days, type I was observed on 26 days, type II on 7 days and type III on 13 days. For 11 out of
31 days, 2 types of profiles were observed on 11 days, and all 3 types on 4 days (Table A4). Occurrence of different
profile types during a single day indicates significant fluctuations of meteorological conditions.

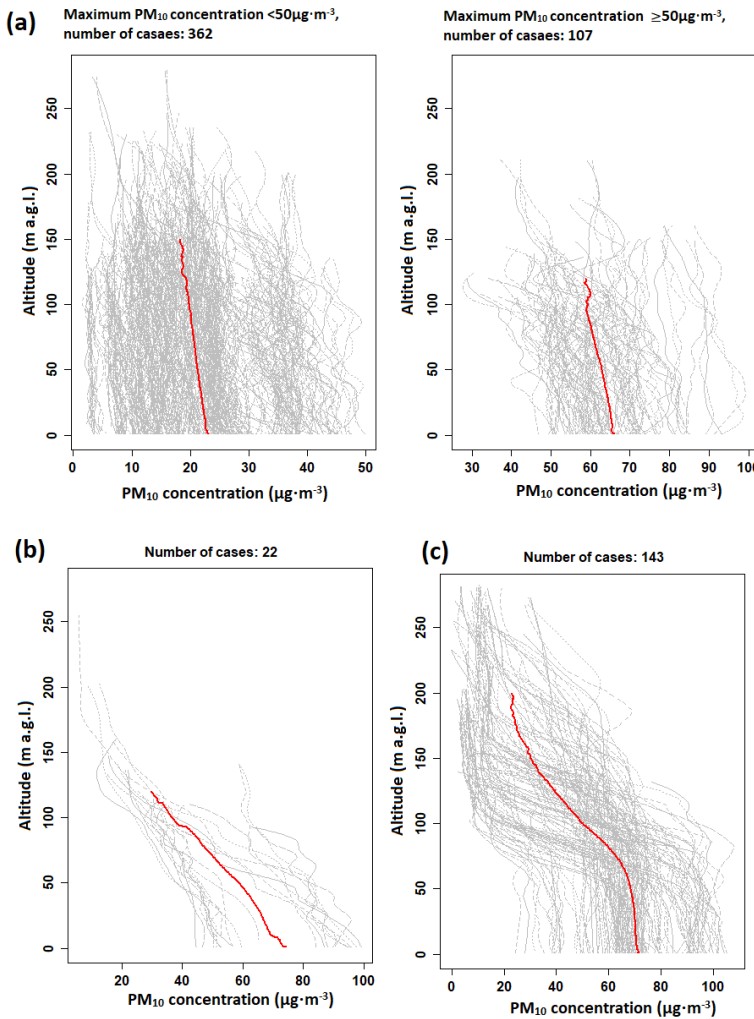

*Figure 3. Classification of PM10 vertical profiles into the three main types: a. type I (it is presented in two plots*
*due to a wide range of PM10 concentration values); b. type II; c. type III*
Explanations: gray lines – individual vertical profiles of PM10 concentration, red lines - mean profiles of a certain
type

Vertical profiles assigned to type III differ a lot in the position and thickness of the transition layer. The dominant
pattern in figure 3c is characterized by a sudden drop in pollution at the valley top which is about 100 m a.g.l. The
transition layer was further determined with the features presented below:
1.   Calculation of mean concentration in the lower layer (up to 70 m above the ground level) and upper layer

(the last 20 m in profile)

2.   Determination of the altitude at which the decrease of $PM_{10}$ concentration in the lower layer is $\geq 15\mu g \cdot m^{-3}$

between two neighbor measurement levels; for the upper layer the difference was set $\geq 5\ \mu g \cdot m^{-3}$. In case

of the occurrence of the transition layer only (i.e. no upper layer), the last point of the profile was

considered the upper level of the transition layer.



3.    Each flight was checked whether it was necessary to manually modify the height of the layers on the basis

of the analysis of the entire profile, and such correction was made for 37 out of 143 profiles.

Figure 4 presents characteristics of transition layer for all selected vertical profiles.

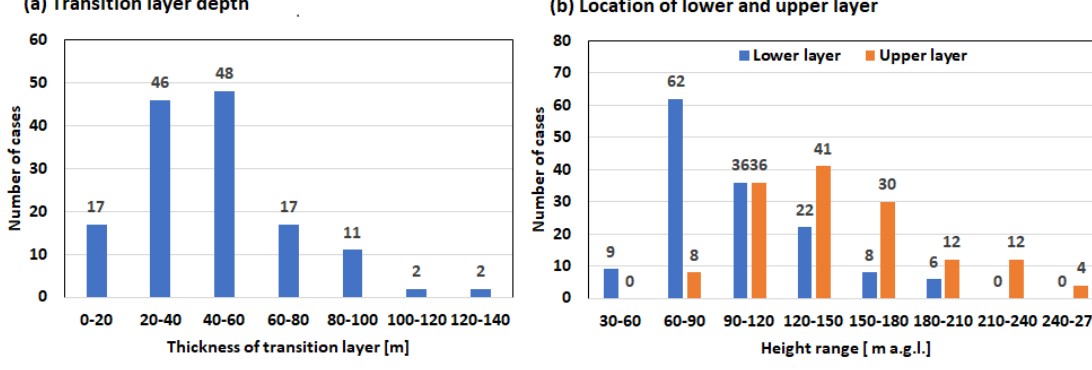


*Figure 4. Characteristics of the* transition *layer in the vertical profiles of PM10 concentrations in type 3.*

It should be noted that the vertical profiles in type I, could have been the lower part of profiles of type III; the low
flight maximum altitude, associated with the occurrence of a strong wind, did not allow to continue the
measurements higher and verify the hypothesis.

### 4.5 Impact of relief and meteorological conditions on $PM_{10}$ concentrations vertical profiles

Type I
On 18 out of 26 days analyzed, mechanical and thermal turbulence led to strong convection. However, the effect
of mechanical turbulence was a quick increase of convection layer thickness during the day, followed with its
sudden decrease in the evening, while thermal turbulence caused gradual development of the convection layer and
its lower thickness. The upper limit of the convection layer was defined with the application of TKE profiles and
reached 300-500 m a.g.l. The flights height on those days did not exceed those values which was the reason of the
almost constant $PM_{10}$ concentration observed.
On 5 out of 26 days analyzed, convection layer was controlled by the thermal turbulence. Its thickness did not
exceed 200 m a.g.l., and wind shear was observed above but the flights reached only 150 m a.g.l. Therefore, the
upper layer with – most probably – much lower $PM_{10}$ concentrations could not be observed. Such scenario is an
example of a modification of the turbulence at the top of CBL, i.e. a reduction of vertical mixing efficiency by
wind shear, presented e.g. in Rodier et al., 2017.

Type II
The sudden decrease of $PM_{10}$ concentration with height in profile type II was an effect of two processes: an increase
of pollutants emission near the ground and removal of the pollution from the upper layers. The latter was due to
mechanical turbulence caused by the presence of the wind shear. The wind shear was the effect of an increase of



wind speed close to the valley top and significant wind direction change caused by the complex topography impact.
Sudden decrease of PM$_{10}$ concentration at 6 out of 7 days was observed at evening hours, after weakening of
convection movements and wind speed close to the ground. During 2 out of 7 days selected, the occurrence of
turbulence was caused by the presence of mountain waves which strongly modified convection movements. The
analysis of the flights showed that it was a short-time phenomenon which can occur during e.g. a momentary lack
of convective movements or a passage of an atmospheric front.
The case study of 27 Jan., 2020, is presented below as an example of the processes described above.  In the early
morning hours until 9 UTC, there was a humid cold pool in the valley, drier and warmer air moved over the valley
from the west. Between 6 and 12 UTC, there was a gradual break of the inversion and a decrease in humidity in
the profile observed at 50 and 100 m a.g.l. at the tower station. Until 12.00 UTC, the PM$_{10}$ concentration at the
ground stations did not change significantly, after 12 UTC an increase of PM$_{10}$ concentration was visible in the
vertical profile. The increased concentration of PM$_{10}$ at Krasińskiego St. compared to other stations maintained
until 17 UTC. The difference in concentration between the ground-level measurement from the balloon point and
Krasińskiego St. was in the range of 50-70 μg·m$^{-3}$ for most of the time. Vertical profiles of TKE indicated that
convection layer during this day reached up to 200-220 m a.g.l., isolines of TKE equal 0.01 m$^2$·s$^{-2}$ and 0.04 m$^2$·s$^{-2}$
are presented at Figure 5c. Flights between 10 and 14 UTC indicated a constant PM$_{10}$ concentration value in the
profile up to 150 m a.g.l. Linear decrease of PM$_{10}$ concentration above 150 m a.g.l. was noticed at higher flights
around 12:30 UTC and 14:00-14:30 UTC. The consequence of the disappearance of convection layer (which began
at 13 UTC) and mechanical pollution removal from the layers above the valley was visible at flights after 14:30
UTC. The strongest decrease in the concentration in the vertical profile was observed during the last flight; the
height of ground layer with stable PM$_{10}$ concentration did not exceed mean height of the buildings in the city (30
m a.g.l.), and above this layer there was a linear decrease in PM$_{10}$ concentration. The decrease in concentration in
the layer up to 150 m a.g.l. was related to the occurrence of a wind direction change from SW to W (see the cross
section at 16 UTC in fig. 5 d); above this layer, a linear increase in wind speed occurred (see wind profiles in fig.
5 c). During the night, there was a separation of the valley wind and topographically channeled airflow, i.e. the
wind in the valley weakened, and at the valley top the wind speed increased.

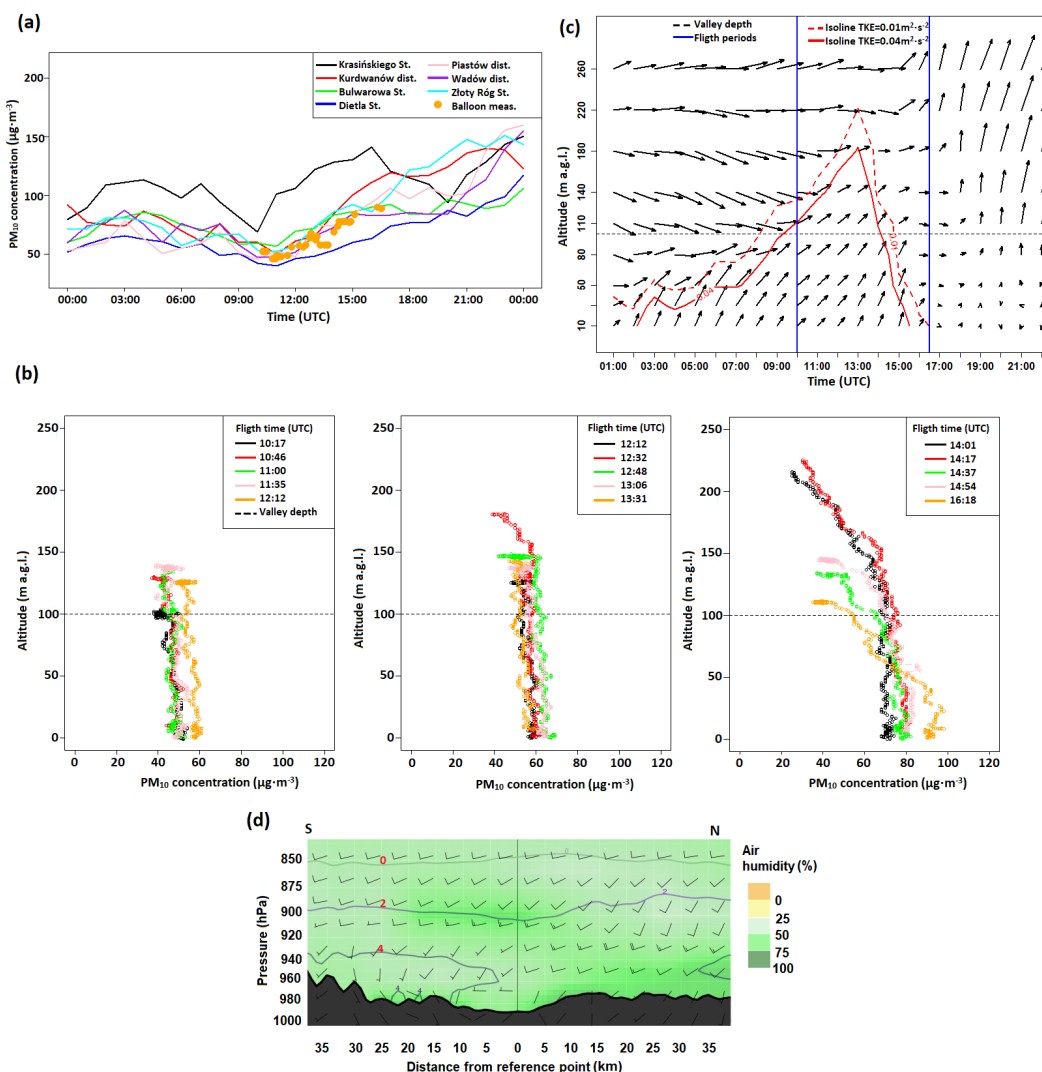

Figure 5. Hourly concertation of PM10 at air pollution stations on 27 Jan., 2020: a) ground-level measurements during balloon soundings, b) vertical profiles of PM10 concertation, c) wind profile forecast with added isolines of TKE equal 0.01 m2 s-2 and 0.04 m2 s-2 for point representing city center. Measurement period is marked with blue vertical lines. (d) Air temperature (contour lines), air humidity (background colour) and wind speed (in knots) and direction (graphical symbols) in the SW-NE cross section through Kraków and its vicinities at 16 UTC for the sounding location.

Explanation: valley depth is the altitude of the hilltops surrounding the valley marked at 100 m a.g.l. with a dashed line in fig. b and c



Type III
Type III of $PM_{10}$ concentration vertical profile was found on more than 40% of measurement days (13
out of 31 days). The vertical wind profiles indicated that during the most of selected days a significant wind
direction change (wind shear) was observed close to the valley top (i.e. about 100 m a.g.l.) or at upper layers. Wind
shear occurred either in a thin layer (i.e. as a sudden change between two neighboring vertical model levels, in a
layer up to 50 m thick), or in a thick layer (100-200 m). The occurrence of the wind shear was also accompanied
by an increase in wind speed, which was responsible for pollution removal from the upper layer. Wind direction
observed at the lower layer was determined by the local topography (valley wind), whereas at upper layer there
was regional topographically channeled airflow. The separation of the two atmospheric layers by a strong wind
shear for selected cases was reinforced by the advection of warmer air (on 8 days out of 13 analyzed). In case of a
cold pool occurrence in the valley, the vertical transport of air pollution was hindered by the thermal inversion
intensification. The analysis of TKE vertical profiles and wind speed showed that the height of the transition layer
depends on the height of the convection layer and the occurrence of wind shear (Fig. 6). The wind shear occurrence
was defined as a wind direction change between two neighbor vertical model levels, and the minimum value was
set to 20 deg. If the predicted height of convection layer and wind shear occurrence occurred at the same model
level, wind shear was connected to jet stream absence which was modifying convection layer. It was observed for
20 flights, on 6 days of 13 analyzed (Fig. 6c).

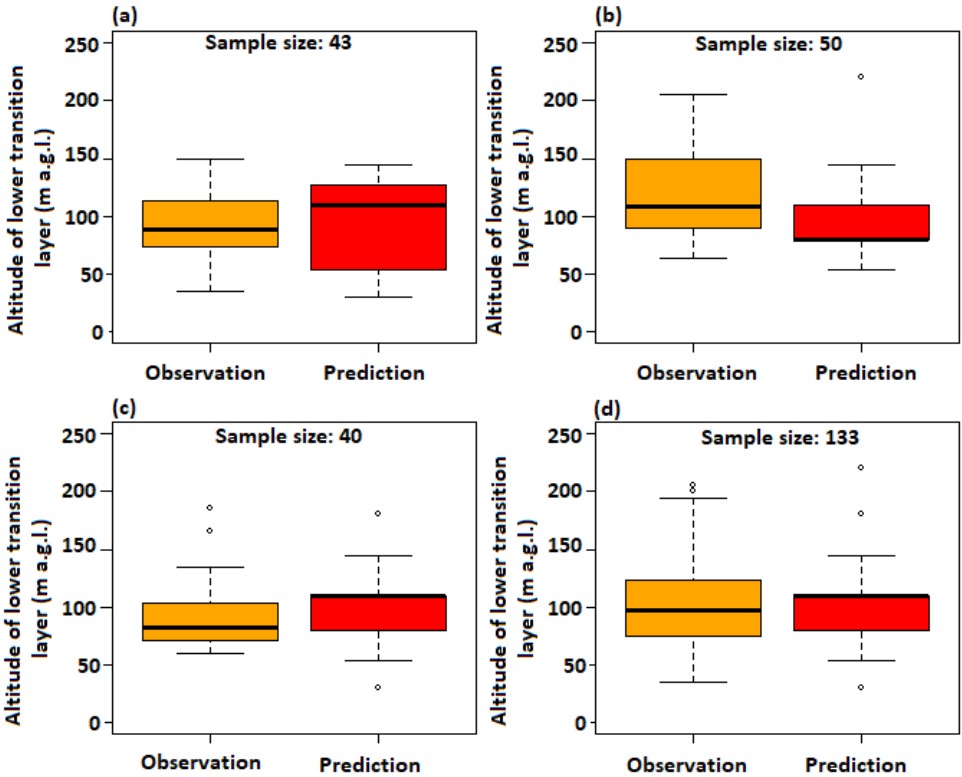


*Figure 6. Boxplots of observed height of lower transition layer and predicted height of the convection layer (a),*
*the wind shear (c), convection layer and wind shear occurrence occurred at the same model level and summary*
*of total cases (d). At figures are included sample size.*

The results presented in Figure 6 indicate that the median for all cases of the altitude of the lower transition layer

oscillates around the valley depth (100 m a.g.l.). The lowest interquartile range of observations is linked with cases

where the dominant factor is the height of convection layer (and the presence of wind shear) (Fig. 6 a and c). The

predicted height of the lower transition layer was the most consistent with the observations for cases where

convection and wind shear occurred at the same vertical model level. For cases where convection height was the

dominant factor, the first quartile is too low and for wind shear the position of the upper quartile and the median

is too low in comparison with the observations.

Data of 28 Nov., 2019, were used as an example of profile type III. Vertical profiles of air humidity and air

temperature from model forecast and measurements from TV tower indicated the presence of a persistent ground

thermal inversion intensified by warm and dry air advection from the south-west (Fig. 7 c-d). The height of the

transition layer did not exceed valley top, and the differences between the individual vertical $PM_{10}$ concentration

profiles were not significant. The height of the transition layer was mostly determined by the height of the

convection layer; wind vertical profiles indicated the occurrence of wind shear above the convection layer. The

limited range of the convection layer at 28 Nov., 2019, was the result of high cloudiness during the daytime. On

that day, foehn conditions were not met at Kasprowy Wierch and Balice station, however the cross section of

AROME CMC 1km model indicated the occurrence of foehn in the south-west Western Carpathians. This
phenomenon could partially contribute to the warm air advection from south-west. Additionally, data from the air
pollution measurement stations showed significant spatial variability of PM$_{10}$ concentration in Kraków. Maximum
hourly PM$_{10}$ concentration difference between measurement points was equal to 170 µg·m$^{-3}$. Ground
measurements at balloon site were similar to those from Piastów dist., and differences between balloon site and
Krasińskiego St. were in the range from 89 to 107 µg·m$^{-3}$.
Similar situations, with significant wind direction change in the vertical profile and weak wind speed, were
presented at e.g. Vergeiner, 2004, Li X. et al., 2012 and Li et al. 2015, for mountain valleys, during hydraulic jump
occurrence. In the upper layer, wind direction is constant while wind speed increases with height.

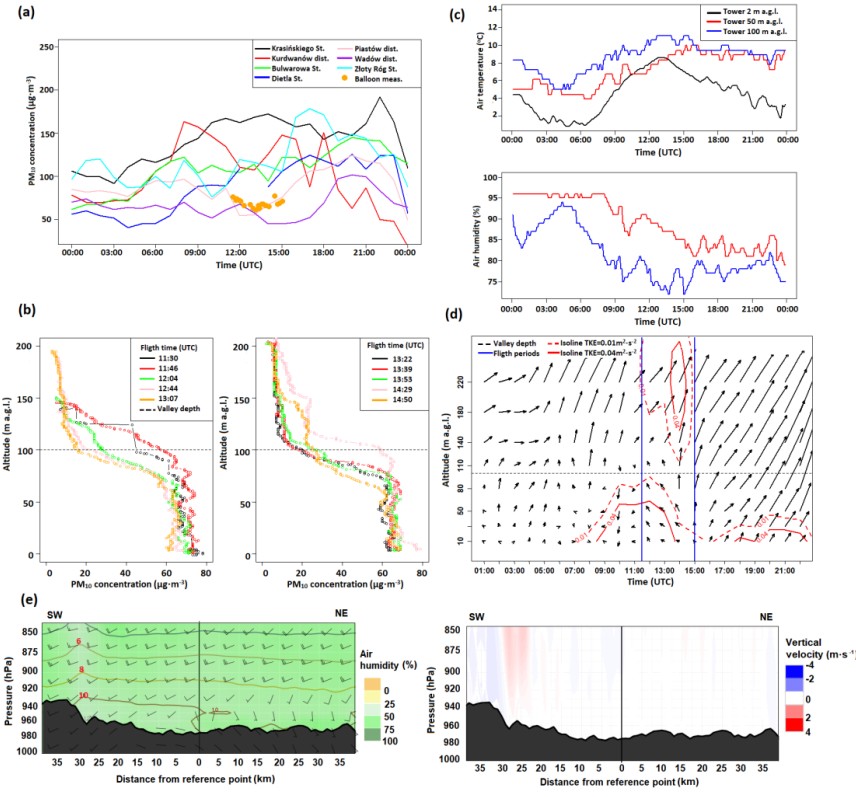

*Figure 7. Hourly concertation at air pollution stations at 28 Nov. 2019 with added ground balloon measurements*
*(a) and vertical profiles of PM$_{10}$ concertation (b), vertical profiles of air temperature (c), wind profile forecast*
*with added isolines of TKE equal 0.01m$^2$·s$^{-2}$ and 0.04m$^2$·s$^{-2}$ for city center with marked measurement campaign*
*period by blue vertical lines (d) and air humidity at 2 levels from TV tower and (e) SW-NE cross section for city*
*center of air temperature (contour lines), air humidity (background), and wind speed (in knots) and direction*
*(graphical symbols), and vertical velocity, at 12 UTC 28 Nov. 2019.*
*Explanation: valley depth is the altitude of the hilltops surrounding the valley marked at 100 m a.g.l. with a dashed*
*line in fig. b and d.*



## 5 Discussion

Studies presenting complex thermal structure of boundary layer (e.g. Xu et al. 2019; Wang et al. 2018) indicate that local pollutants are mostly trapped in the lowest layer. The occurrence of multi-layer vertical structure in the boundary layer were noticed during the foehn periods, too, where warm air advection caused the intensification of the air temperature inversion and CAP, and reduction of the available air volume for mixing the pollutants (e.g. sandwich foehn occurrence: Li X. et al., 2015; Vergeiner, 2004). In the present paper, for the days with balloon flights, the occurrence of $PM_{10}$ profile type III was connected with the advection of air masses from the south. Such advection direction may be linked to the foehn wind occurrence in the Tatra Mts. Therefore, it was checked whether such advection is linked to high $PM_{10}$ concentration differences between the measurement points within the city, especially between the western, narrow part of the valley and the eastern, wide part. For both cold seasons, cases of $PM_{10}$ concentration differences $> 50$ µg m$^{-3}$ which lasted at least for 5 hours constituted 10.9% of the study period. For half of the cases, the dominating wind direction noted in Libertów was from the sector 130-270°. In both cold seasons, wind direction from the sector 130-270° was noted in 52.6% of cases, which shows that it is an important factor controlling $PM_{10}$ spatial patterns, but the impact is diversified.

Research presenting impact of PBL dynamics, confirms that during convective conditions (mechanical and thermal turbulence) vertical distribution of PM concentrations is uniform (Li et al. 2019; Strbova et al., 2017; Wang et al. 2018). Mechanical turbulence can be caused by strong wind shear connected to LLJ (Li et al. 2019), mountain waves (Zangl, 2003), hydraulic jump (Kishcha et al. 2017), rotors (Kunin et al. 2019) or passage of an atmospheric front. In the present study, wind shear turned out to be the most important factor in terms of $PM_{10}$ vertical profile modification. In the case of the study area under investigation, the wind shear is generated due to the relief impact, i.e. the presence of a large valley, blocked on one side with the hills. Studies presented in Sheridan (2019), indicate that the valley width is an important parameter affecting the interactions between CAP and air flow above the valley. For valleys which depth exceeds the depth-scale of the nocturnal stable boundary layer, processes related to daytime insolation may be not strong enough to break the cold-air pool.

The data used included both measurement and model data which allowed to verify, as much as possible, the numerical weather predictions. Prognosis of e.g. wind field and TKE is highly dependent on the inclusion of various topographical features in the model formula. Local-scale phenomena like low level jet, cold pool occurrence, and katabatic flows are often under-represented in the model analysis, so the verification with observations is needed.

The meteorological and $PM_{10}$ data for the study periods were compared to the data for the whole two cold seasons and it was found out that they are representative for the situations with very low wind speed and higher than usual air pollution. Therefore, the analyses' outcomes are valid for those periods within the cold season when the aerosanitary conditions are the worst. Additionally, the results obtained may be considered as representative for cities located in large river valleys of Central Europe and applied in the studies concerning the air quality there.

## 6 Conclusion

The results of our study present how the wind shear generated in a local scale by the diversified relief's impact can be a factor which might significantly modify the spatial pattern of $PM_{10}$ concentration. We focused mainly on


the events characterized by high surface-level $PM_{10}$ concentrations in the city centre, as such situations are the
most dangerous and the most important from the point of view of the inhabitants' health. High $PM_{10}$ concentrations
are usually linked to low wind speed occurrence, and all $PM_{10}$ concentration vertical profiles were obtained in such
conditions, due to safety regulations concerning the balloon operation. The flights' height depended on the height
at which the wind speed was too high to continue the uplift. Vertical profiles of $PM_{10}$ concentration are also
strongly dependent on the thickness of the convective layer. We have distinguished three main types of $PM_{10}$
concentration vertical profiles, with type II being the least numerous and observed sporadically, usually as an
intermediate short-term form occurring during the development of either type I, or type III. In fact, the air layer
inside the valley with constant high $PM_{10}$ values of vertical concentrations described as type I, was usually found
to be only a lowermost section of type III, but the whole profile could not be observed as the wind speed at higher
levels was too high to continue the flight. Type III presents the situation where the impact of the wind shear on
$PM_{10}$ concentration profile is not linked mainly to the change in wind speed, like in type I, but to the change in
wind direction; the wind speed had to remain low within the whole profile as otherwise the balloon flight could
not be realized. In type III, the sudden decrease in $PM_{10}$ concentrations above the layer with its high constant
values are due to the advection of different air masses in a regional scale. The analysis of $PM_{10}$ profiles from all
flights allows to distinguish three vertical zones of potential air pollution hazard within the valley (about 100 m
deep) and the city of Kraków:
1. up to about 60 m a.g.l. – the zone where during periods of low wind speed, air pollution is potentially the

highest and the duration of such high levels is the longest, i.e. the zone with the worst aerosanitary

conditions;

2. about 60-100 m a.g.l. – transitional zone where the large decrease of $PM_{10}$ levels with height is observed;
3. above 100-120 m a.g.l. – the zone where air quality is significantly better than in the zone 1, either due

to the increase of the wind speed, or due to the wind direction change and advection of different, clean

air masses

Further research is planned, including night balloon measurements during high $PM_{10}$ concentration episodes.
Additionally, it is planned to determine the share of particles of various size fractions in the air pollution with the
sensors where light scattering method is applied.



**APPENDICIES**
*Table A1. Height of the lowest 87 model vertical levels (v.l.) up to 3 km of altitude, used in forecast.*

| No. of v.l. | Height of v.l. (km a.g.l.) | No. of v.l. (cont.) | Height of v.l. (km a.g.l.) |
|---|---|---|---|
| 1 | 0.009 | 20 | 0.969 |
| 2 | 0.030 | 21 | 1.055 |
| 3 | 0.053 | 22 | 1.144 |
| 4 | 0.079 | 23 | 1.237 |
| 5 | 0.110 | 24 | 1.334 |
| 6 | 0.143 | 25 | 1.435 |
| 7 | 0.180 | 26 | 1.537 |
| 8 | 0.221 | 27 | 1.640 |
| 9 | 0.264 | 28 | 1.744 |
| 10 | 0.311 | 29 | 1.849 |
| 11 | 0.362 | 30 | 1.957 |
| 12 | 0.415 | 31 | 2.066 |
| 13 | 0.472 | 32 | 2.178 |
| 14 | 0.533 | 33 | 2.292 |
| 15 | 0.597 | 34 | 2.408 |
| 16 | 0.664 | 35 | 2.527 |
| 17 | 0.735 | 36 | 2.649 |
| 18 | 0.809 | 37 | 2.773 |
| 19 | 0.887 | 38 | 2.900 |



















*Table A2. Physics schemes used in AROME CMC 1 km model.*

| | |
|---|---|
| Dynamics | Nonhydrostatic ALADIN (Benard et al., 2010) |
| Turbulence | Prognostic turbulent kinetic energy (TKE) combined with diagnostic nixing length (Cuxart et al., 2000;Bougeault and Lacarrere, 1989) |
| Radiation | Longwave Rapid Radiative Transfer Model (RRTM) radiation scheme, Morcrette shortwave radiation scheme from European Centre for Medium-Range Weather Forecasts (ECMWF) |
| Microphysics | Three-class parameterization (ICE3) |
| Shallow convection | Pergaud, J., Masson, V., Malardel, S., and Couvreux, F., 2009 (PMMC09) (Pergaud et al., 2009) |
| Deep Convection | - |
| Clouds | Statistical cloud scheme |
| Surface scheme | SURFEX (Masson et al., 2013) |


*Figure A1. Orography map of AROME model domain with resolution 1 km x 1 km.*

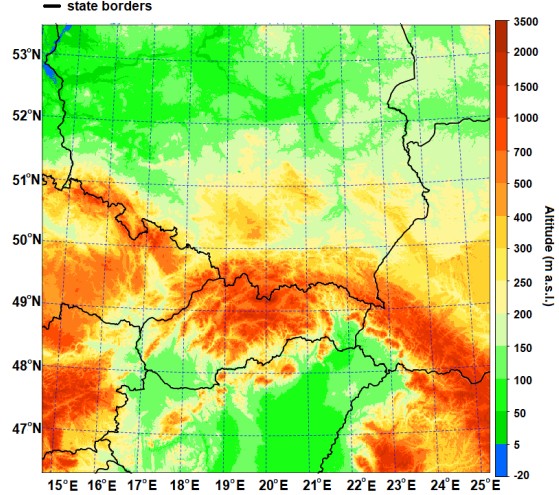






*Figure A2. Wind rose for three stations located in the valley Balice (a), Reymonta St.(b), Igołomia (c) and one at*
*the nearest hilltop station Libertów (d) for cold seasons 2018-2020.*

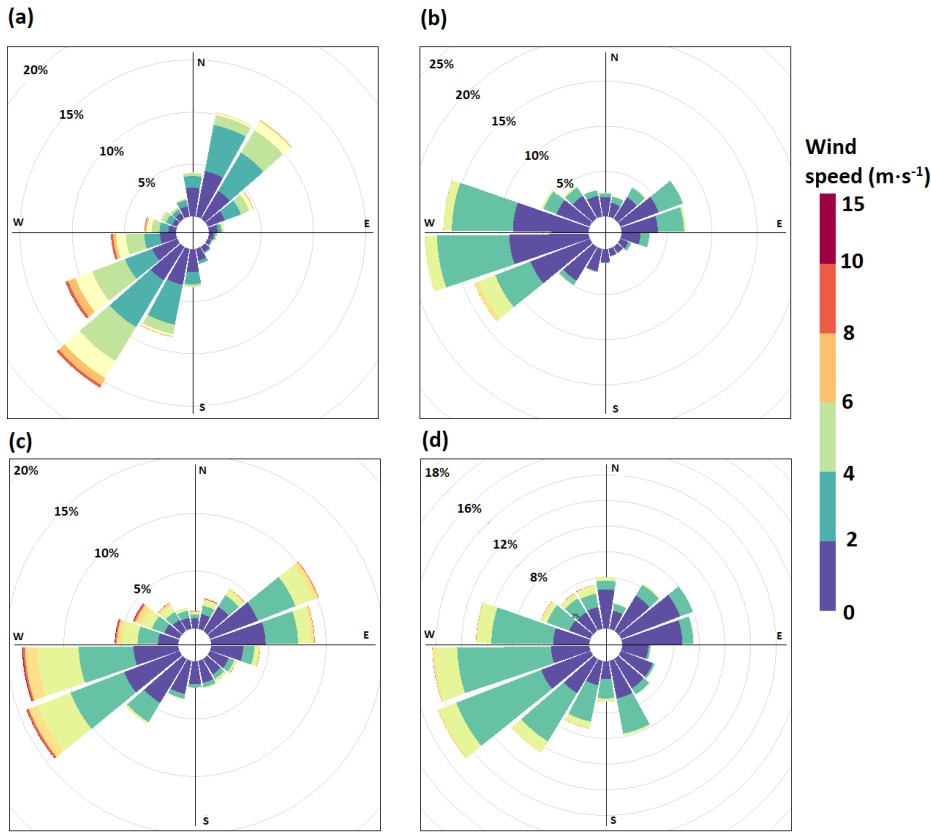




*Figure A3. Analysis of hourly PM10 concentration at air pollution stations in Kraków compared to wind speed*
*from Reymonta St. station: a) Krasińkiego St., b) Dietla St., c) Bulwarowa St., d) Złoty Róg St., e) Kurdwanów*
*dist., f) Piastów dist., g) Wadów dist.  To presented data is fitted logarithmic curve, at right corner is included*
*curve equation.*

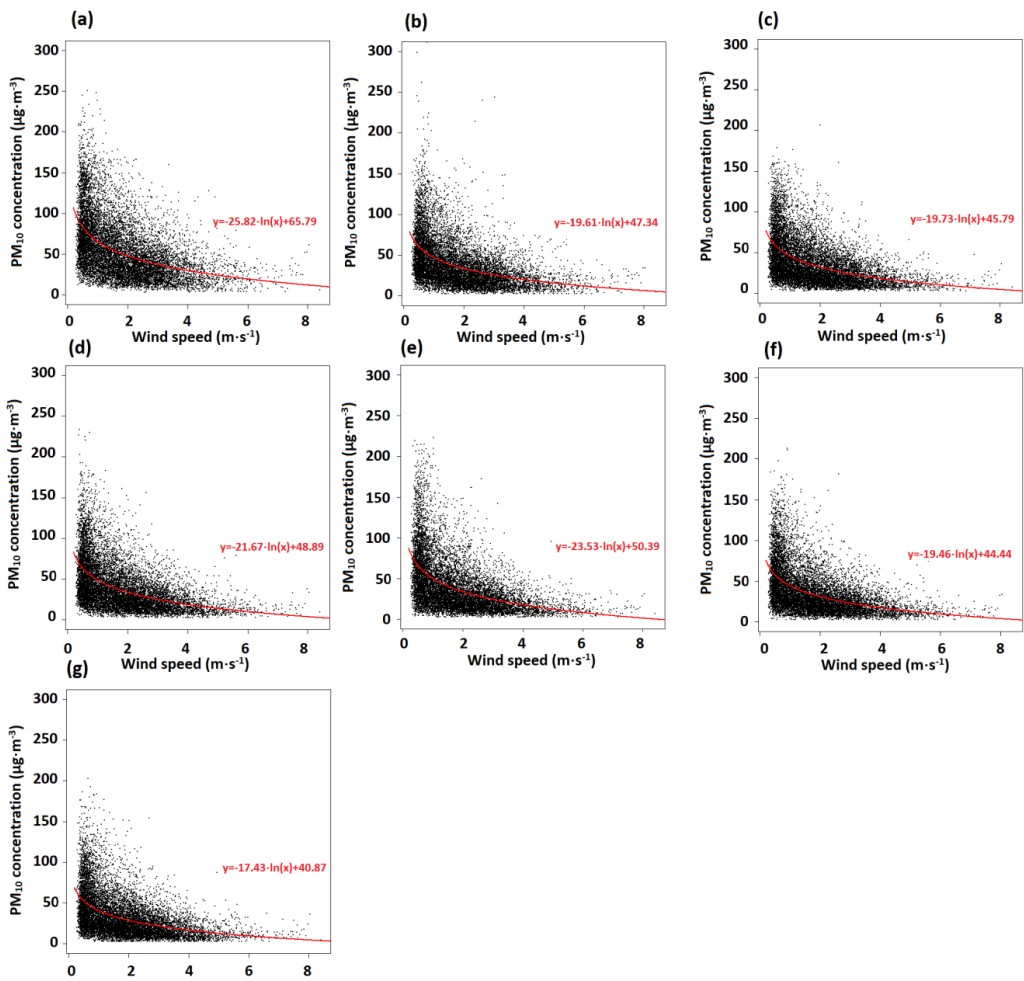









*Table A3. Distribution of the temperature gradient between levels 200 and 50 m a.g.l. depending on the wind*
*speed at a height of 50 m a.g.l. for city center at two cold seasons 2018-2020 obtained from AROME model*
*forecast.*

| | | Wind speed range at 50 m a.g.l. [m·s⁻¹] | | | | | |
|---|---|---|---|---|---|---|---|
| | | [0;2) | [2;4) | [4;6) | [6;8) | [8;10) | [10;20) |
| Air temperature gradient range between layers 200 and 50 m a.g.l. [°C/100m] | [-1.5;-1.0) | 371 | 649 | 689 | 437 | 171 | 61 |
| | [-1.0;-0.5) | 404 | 965 | 1065 | 900 | 352 | 171 |
| | [-0.5;0) | 244 | 634 | 429 | 145 | 23 | 4 |
| | [0;0.5) | 306 | 625 | 283 | 41 | 3 | 0 |
| | [0.5;1) | 322 | 445 | 112 | 6 | 1 | 0 |
| | [1;1.5) | 303 | 309 | 65 | 4 | 2 | 0 |
| | [1.5;2) | 193 | 190 | 34 | 7 | 0 | 0 |
| | [2;5) | 266 | 316 | 53 | 5 | 2 | 0 |
| | [5;10) | 2 | 31 | 0 | 0 | 0 | 0 |




*Table A4. List of measurement campaign with specified PM10 profile observed during selected day.*

| No. | Type I<br>26 days (10 days with PM10 maximum concentration above 50μg·m⁻³, marked with **text in bold**) | Type II<br>7 days | Type III<br>13 days |
|---|---|---|---|
| 1 | | | 28.11.2019 |
| 2 | 01.12.2019 | | |
| 3 | **05.12.2019** | | |
| 4 | 06.12.2019 | | |
| 5 | 09.12.2019 | | |
| 6 | **11.12.2019** | | |
| 7 | | | 12.12.2019 |
| 8 | 13.12.2019 | | |
| 9 | | 17.12.2019 | 17.12.2019 |
| 10 | **19.12.2019** | | 19.12.2019 |
| 11 | 21.12.2019 | | |
| 12 | 22.12.2019 | | |
| 13 | **02.01.2020** | 02.01.2020 | 02.01.2020 |
| 14 | **03.01.2020** | 03.01.2020 | |
| 15 | 06.01.2020 | | |
| 16 | 07.01.2020 | | 07.01.2020 |
| 17 | **09.01.2020** | | 09.01.2020 |
| 18 | 12.01.2020 | | |
| 19 | 13.01.2020 | | |
| 20 | | | 14.01.2020 |
| 21 | **16.01.2020** | | 16.01.2020 |
| 22 | 20.01.2020 | | |
| 23 | 25.01.2020 | | |
| 24 | **26.01.2020** | | 26.01.2020 |
| 25 | **27.01.2020** | 27.01.2020 | 27.01.2020 |
| 26 | | 28.01.2020 | |
| 27 | 15.02.2020 | 15.02.2020 | 15.02.2020 |
| 28 | 17.02.2020 | | |
| 29 | **20.02.2020** | 20.02.2020 | 20.02.2020 |
| 30 | 01.03.2020 | | |
| 31 | 03.03.2020 | | |









Code availability: not applicable
Data availability: not applicable
Author contribution: Piotr Sekuła: Conceptualization, Methodology, Validation, Formal analysis , Visualization,
Writing - Original Draft, Writing - Review & Editing, Anita Bokwa: Conceptualization, Methodology, Formal
analysis, Writing - Original Draft, Writing - Review & Editing, Jakub Bartyzel: Conceptualization,
Methodology, Investigation, Writing - Review & Editing, Bogdan Bochenek: Conceptualization, Writing -
Original Draft, Łukasz Chmura: Conceptualization, Investigation, Resources, Writing – Review & Editing,
Michał Gałkowski: Conceptualization, Investigation, Resources, Writing - Review & Editing, Writing - Review
& Editing, Mirosław Zimnoch: Conceptualization, Methodology, Investigation, Writing - Original Draft, Writing
- Review & Editing
Competing interests: The authors declare that they have no conflict of interest

**Acknowledgements:** The authors wish to thank Balon Widokowy sp. z o.o. for providing a tethered balloon for
the measurement of PM10 vertical profiles in Kraków. This research was partly funded by the EU Project
POWR.03.02.00-00-I004/16 (PS) and Ministry of Science and Higher Education subsidy, project no.
16.16.220.842-B02.

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
