# Peer review of "Measurement report: Effect of wind shear on $PM_{10}$ concentration vertical structure in urban boundary layer in a complex terrain"

_Atmospheric Chemistry and Physics, 2021_

## Author Response (AR1)

**Response to the Referees**

Title: Measurement report: Effect of wind shear on PM10 concentration vertical structure in urban boundary layer in a complex terrain Author(s): Piotr Sekuła et al. MS No.: acp-2021-93

Dear Sirs,

We are grateful to both anonymous reviewers for their detailed and constructive comments on the content of the manuscript. The comments allowed us to improve significantly the overall quality of the manuscript and helped to remove several minor shortcomings.

With best regards,

Anita Bokwa

**Reviewer 1**

The work was prepared by a team of good specialists in the field of climatology and meteorology, statistical methods and atmospheric pollution. Measurements of the vertical structure of PM10 concentration and temperature in an atmosphere layer >200m in Krakow were taken during two winter seasons thanks to the use of sightseeing balloon. The authors also used all available meteorological and air pollution data from all measuring stations in Krakow and from meteorological models to explain the vertical structure of PM10 concentration. The results were discussed with other similar measurements around the world. The authors have demonstrated knowledge of the latest literature on this subject. This is evidenced by the citation of the latest works, even from 2020 and 2021, for example lines: 37, 44, 50, 60-62 71, 80-81, 84).

The most important result of the research is the finding of a significant influence of wind shear on the vertical distribution of PM10 concentration and the determination of three vertical zones of air pollution hazards. These results are representative of other parts of Europe with poor aero sanitary conditions.

In terms of content, I fully accept the text of the publication. If possible, I suggest making a few technical corrections, which are listed below.

Technical corrections:

Line 108: (PM10 emissions... 2020). In the reference is the year 2021, see line825.

Answer:

**Reference has been modified.**

Line 1003: change (Raport... 2020) to (Roczna... 2020). See line 834.

**Answer:**

**Reference has been modified.**

Line 122: (Kraków, 2019)? Is this citation?

**Answer:**

**Reference has been modified.**

Table 1 and 2: add "°" for Lat and Lon: °N, °E

**Answer:**

**Missing symbols have been added.**

Lines 398 and 399: 2-4 m  $\cdot$  s-1 is to be written together on one line.

**The text has been modified.**

Lines 552, 556, 566-567: citations must be in chronological order.

**Answer:**

Reference has been modified.

Line 568: Zangl - in reference list, line 892 is Zängl.

**Answer:**

Reference has been modified.

Lines 739-740: This position is not cited in the text (Drechsel and Mayr, 2008).

**Answer:**

Reference has been added to the text.

Lines 834-837: Roczna... move to line 843.

**Answer:**

Reference has been modified.

**Reviewer 2**

Here the authors present data collected from surface stations and balloon measurements in and around the city of Krakow, Poland. Through comparison of PM10 measurement patterns, along with model output, the authors attempt to explain observed differences in observed PM10 vertical profiles through variability in atmospheric dynamics, including wind shear

strength. The use of commercial balloon measurements in this way seems to be an interesting and useful source of atmospheric data, and I think it deserves recognition and dissemination. It is also clear that the authors have compiled a great deal of observational data and model output in their exploration of pollution in this region, and it is likely to be useful for many locations with similar meteorological and topographical features. That said, I have a number of concerns regarding the quality and types of analyses being performed that I would like to see addressed before publication.

**General Comments**

Overall Organization - The total quantity of information, visualizations, and explanation is quite high relative to the scope of data presented. However, in its current form it seems inefficient and scattered to my eye. A great deal of written space is dedicated to relatively straightforward topics (e.g. regional topography, various model configurations), while other analytical choices of great importance (e.g. vertical profile categorization and statistical analysis of potential influences) are only described briefly in passing. I would expect to see significant streamlining and focusing of the primary narrative in this paper before I would consider it ready for publication.

**Answer:**

We believe that it is necessary to provide some background information in the first part of the paper because the area of southern Poland may be not well known to many readers. Vertical profile categorization has been modified (see one of the answers below), and also the statistical analysis of potential influence of atmospheric phenomena (wind shear and convection layer height) on vertical profile of PM10 concentration has been enlarged.

Clarity of Text and Visualizations - Figure quality varies greatly, and some in particular do not seem to directly support the claims made about them. Phrasing and word choice is sometimes awkward and difficult to parse. As one example, "wind direction change" is sometimes used in a way that makes it hard to tell if the authors are referring to a change over time, or a change across the vertical profile (wind shear). More attention to clarity and precision would be appreciated throughout.

**Answer:**

**Figures 5-7 have been modified to make it easier for readers to understand. Text has been modified, with particular attention to sentences wind direction change and wind shear.**

Analysis Choices - Several key steps taken in the analysis of observations seem to lack reproducibility and objectivity, raising concerns in my mind regarding the robustness and validity of the conclusions drawn from them. In particular, the selection and sorting of vertical profiles, as well as the subsequent evaluation of associated meteorological conditions deserves more description and potentially some rethinking in approach. Specifically:

 The authors state that they use a "subjective method of fitting the linear curve to each vertical profile" based on "R squared coefficient, the angle of the straight line and residual values classification". They further clarify that objective classification methodologies could not be used "due to differences in flight heights and the PM10 measurement altitudes". With all of the classification approaches available, I find this assertion difficult to accept, especially if the classification is being performed on quantitative metrics like the ones listed. Without a clearer, more objective approach I find it hard to consider this procedure sufficiently robust and reproducible.

**Answer:**

Vertical profile categorization has been modified, for each vertical profile of PM10 the sigmoid curve has been fitted, separately. The logistic curve which was used in the study is determined by equation:

**Y=c+(d-c)/(1+exp(b(X-e)))**

where: b – slope around the inflection point; c - lower asymptote; d - higher asymptote; e - X value producing a response half-way between d and c.

The parameter b can be positive or negative and, consequently, Y may increase or decrease as X increases. At the first step, all possible parameters b, c, d, e were determined, if the lower asymptote c was below 0, fitting curve was repeated with default value of parameter c equal to 0 (minimum PM10 concentration in atmosphere).

Additionally, to better analyze fitted S-shape curve, at the region of inflection point a linear curve was fitted to determine the intersection with the asymptotes c and d (variables y1 and y2). Differences between y1 and y2 represented transition layer depth.

In aim to separate vertical profiles into three groups, boundary conditions were determined based on the obtained parameters and their statistical features.

- Group I: PM10 concentration at ground layer (i.e. below 10 m a.g.l.) was lower than 30µg·m-3 (275 vertical profiles) or difference between PM10 concentration at the ground layer and in the upper layer (close to the maximum flight altitude) was less than 25µg·m-3 (208 vertical profiles).

- Group II: difference between PM10 concentration at the ground layer and in the upper layer was greater than 25µg·m-3 and variable y2 is in range [-200;30] m a.g.l. (determined experimentally for this data set) (17 vertical profiles)

- Group III: difference between PM10 concentration at the ground layer and in the upper layer is greater than  $25\mu g \cdot m^{-3}$  and variable y2 is greater than 30 m a.g.l. (134 vertical profiles)

• Following this subjective sorting, some comparison of local conditions for each profile is performed and described. However, the figures and metrics presented to support the final conclusions are not sufficiently clear and convincing, to my eye. Figure 6, in particular, would seem in the text to be a key figure in supporting final conclusions, but after several readings I still cannot figure out what it is supposed to be showing. Figure 7 highlights a particular day chosen as an example, but the information is scattered, unclear, and leaves many questions unanswered regarding the representativeness of the chosen day and the reasons for not somehow showing the characteristics of ALL observed days, rather than one or two. Throughout the final pages of text, I kept looking for the conclusive evidence that succinctly and unambiguously showed the connection (with uncertainties and other statistical metrics) between the patterns being studied, but I was not able to find this. If such evidence is in there, it should be made more clear. If not, it must be added.

**Answer:**

Figure 6 has been modified to better present significance of wind shear and height of convection layer on vertical distribution of PM10. Figure 7 has been modified to better meteorological conditions on selected day.

In order to confirm the representativeness of the presented day for the entire period, a paragraph describing the characteristics of all days when the profile type 3 was observed was added to the article

• Finally, I would like to suggest that the order of operations being performed here (first manually sort vertical profiles, then try to analyze atmospheric conditions) is making the authors' (and reader's) task more difficult than it ought to be. If the main thesis is that strong wind shear drives key differences in surface and vertical profile PM10 patterns, why not first sort days based on this unambiguous metric (strength of wind shear) and then compare PM10 levels, boundary layer depth, vertical profiles, or any other dependent property based on that? Either way, the connection between pollution results and their potential drivers needs to be explored in a clearer, more defensible manner.

**Answer:**

The manual method of sorting vertical atmospheric profiles has been replaced by the objective method. Statistical analysis of the significance of selected factors (wind shear and convection layer height) on the position of the transition layer in PM10 vertical profile was determined using the root mean square error, Pearson correlation coefficient and p-value calculated from Mann – Whitney U test.

**Minor Comments**

• References: Several references are in a notation unfamiliar to me, for example (Air... 2020). Is this a placeholder? I think these should be standardized and revised, unless I am missing something.

**Answer:**

**The references have been modified accordingly the suggestion.**

• Figure 1: The maps are useful, but their distance from corresponding tables makes it hard to interpret their labels. This may be better resolved at a later stage of typesetting, but some effort to line these up in some way would be helpful.

**Answer:**

Figure 1 has been divided into two figures in order to facilitate the reader analysis of the studied region and the location of measurement points in this area.

 Flight frequency details: What were the patterns of balloon flights over the course of each day? How much of the vertical profile patterns observed can be attributed to regular diurnal cycles? If the "frequency of flights depended on meteorological conditions" is it possible that some sampling bias has been introduced based on these limitations, and if so how is it being addressed?

**Answer:**

The Figure presenting the frequency of flights and the maximum flight altitude has been added to the Appendix. Information concerning unfavorable meteorological conditions has been added to the paragraph.

• Figures 5 and 7: These panel plots appear to contain a great deal of information, but they are overpacked and underexplained. It's unclear in their current form what they are trying to show, on the whole, and some presentation details significantly impede clarity. For example, the humidity color bars run from 0 to 100%, but the data appear

to be uniformly in one small sliver of that space, making the entire plot uniformly green. Both of these figures need rethinking.

**Answer:**

**Both figures have been modified according to the suggestion.**

• Figure 6: As mentioned, this figure is very confusing and it's unclear what the takeaway message is supposed to be. Furthermore, panel (b) appears to never be described in the caption.

**Answer:**

Figure 6 has been modified, also figure caption has been corrected.